# Compound Optimization of Territorial Spatial Structure and Layout at the City Scale from “Production–Living–Ecological” Perspectives

**DOI:** 10.3390/ijerph20010495

**Published:** 2022-12-28

**Authors:** Menglin Ou, Jingye Li, Xin Fan, Jian Gong

**Affiliations:** 1Department of Land Resource Management, School of Public Administration, China University of Geosciences, Wuhan 430074, China; 2Center for Turkmenistan Studies, China University of Geosciences, Wuhan 430074, China

**Keywords:** MOP (Multi-Objective Programming), FLUS (Future Land-Use Simulation), coupling model, pattern optimization, Wuhan

## Abstract

Land-use optimization, as an important resource-allocation method, can be defined as the process of allocating various activities to different geographic units. How to manage and control land expansion has become an urgent issue, leading a series of problems such as environmental damage and a sharp decrease in cultivated land, leading to unfavorable phenomena such as excessive urban expansion, occupation of cultivated land and important ecological spaces, and overheating of real estate development. Based on the land-use data of Wuhan city in 2020, a coupling MOP (Multi-Objective Programming) and FLUS (Future Land-Use Simulation) model was used to examine the national spatial structure and the optimization of the spatial layout. Our results showed that (1) in terms of quantitative optimal allocation, the ecological space and urban space increased, while the agricultural space greatly decreased under the three development scenarios. (2) In the simulation of spatial layout, the urban space mainly expanded vertically in the north–south direction. In the ecological space scenario, the ecological space occupied part of the cultivated land in the northeast of the city, resulting in a high degree of landscape fragmentation, which is not conducive to large-scale agricultural management. However, under optimal comprehensive benefit, part of the fragmented ecological space in the western part of Wuhan was transformed into an agricultural space. (3) A combination of the MOP and FLUS models could effectively determine land-use structure and address spatial layout optimization problems and can project space in the future urban resource configuration mode. This finding can provide a reference for the optimization of the spatial structure and layout of similar cities.

## 1. Introduction

Since its reform and opening, China’s comprehensive national strength has increased significantly, and its development drew attention worldwide [1,2,3]. However, following the vigorous urban construction demand of the rapid economic development [4,5,6], the urban space within the city continues to expand [7,8,9], the agricultural and ecological space are constantly compressed [10,11], and the ratio of the three spatial structures gradually tends to be unbalanced [2,12,13,14,15,16]. This phenomenon directly leads to the deterioration of the ecological environment and the waste of resources among other problems, thereby intensifying the contradiction between economic development, resources, and environment, and posing a serious threat to the sustainable development of national land space. In 2020, the Fifth Plenary Session of the 19th Central Committee of the Communist Party of China proposed that in the new development period, we should rapidly optimize the layout of national land and space, build a “new pattern of development and protection of national space” and a “national space layout and support system for high-quality development” [17,18,19,20,21]. Therefore, the scientific and accurate optimization of the layout of territorial space is one of the major and urgent concerns in regional development [2,7,17,22,23,24,25,26,27,28]. To optimize the land space layout, we need to solve the problem of land-use layout optimization first. Most of the land resource optimization allocation models used by domestic and foreign scholars in the early stage are limited to the optimization of the quantitative structure of land use. The MOP (Multi-Objective Programming) model was used to optimize the quantitative structure of land resources, and then considered the quantitative optimization result as the constraint condition of Cellular Automata (CA) simulation to optimize the spatial layout [25,29,30,31,32]. The effective unification of the optimization of land resources quantity structure and spatial layout was realized. In recent years, numerous local and international scholars have used other models to optimize land-use quantity and layout. The future land-use demand was estimated through grey prediction and an objective optimization model, and realized land-use layout optimization based on the CLUE-S model [20,33]. A land-use pattern optimization model was established by combining multi-agent (MA) and particle swarm optimization (PSO) [34,35]. And the compound optimization of land-use structure and layout at the county level was proposed by integrating the MOP and FLUS model [35,36,37].

Studies have shown that the optimization scheme combined with MOP and Land-Use Simulation model is superior to the single-model optimization scheme. In addition, the Future Land-Use Simulation (FLUS) model was developed based on the traditional cellular automata model [8]. The simulation accuracy is higher than that of CLUE-S and ANN-CA [38]. However, while solving the multi-objective optimization model of land use, most scholars at this stage adopt the subjective weighting method to determine the weight of the sub-objective function in the overall system, and the optimization results obtained cannot be guaranteed to be scientific [35,39,40,41]. In addition, most scholars only focus on optimizing the allocation mode of land-use resources at the present stage. In the new period of national promotion of “integration of multiple plans” and construction of territorial space planning system, exploring the layout optimization method from the perspective of spatial integrity is of significant practical value [40,42].

The optimal allocation of land resources is a complex, multi-level, and multi-objective system engineering, which requires continuous fitting and decision-making. The optimization content mainly includes two aspects, one being the allocation model that optimizes the macroscopic land-use quantity structure, and the other being the allocation model that optimizes the microscopic land-use spatial pattern, to improve the sustainable utilization rate of land resources and maintain the relative balance of the land environment and ecosystem. However, in most developing countries, following the acceleration of urbanization, the land-use structure that puts economic growth in the first place will damage the environmental ecological function of land resources. In certain rapidly developing areas, the government often resorted to administrative measures to interfere with the allocation of land resources and thereby protect the ecological and environmental resources of the region. In view of this, many scholars have studied the optimization of land resource allocation. Certain scholars combined the linear programming model as a tool of geospatial modeling with GIS, and considered the lowest rural unemployment rate as the objective function to study the Mediterranean coastal areas of Spain, to optimize land use. However, the linear programming model addresses nonlinear problems and the interaction between land-use types; therefore, it is not applicable for complex optimization problems of land use. In addition to the weights for each target given, it tends to join the planners of subjective judgment and reduce the optimization. In this case, the heuristic algorithm was integrated into optimization of land use, such as the simulated annealing, genetic, ant colony, and particle swarm optimization algorithms. The genetic algorithm has drawn considerable attention and has been used extensively. Presently, the optimal allocation of land resources mainly focuses on the optimization of land-use structure or spatial pattern, and research studies on the combination of the two are relatively limited. Despite the related research conducted, practical and effective technical methods are lacking.

Therefore, this study introduced the improved non-inferior hierarchical genetic algorithm (NSGA II) in the process of solving the MOP model of land use, which effectively avoided the subjectivity caused by the traditional weighting method to deal with the multi-objective optimization function and yielded more scientific and accurate optimization results [2,43]. At the same time, based on the number of optimization of land use, the FLUS model was applied to implement the layout optimization of land use, and try to “three types of” technical code for space and based on “three” classification standards for the investigation of cohesion, in Wuhan city, Hubei province as the research object, analyzing scales in Xiamen city: urban space, agricultural space, and ecological space layout optimization methods. This study aimed to provide a scientific method for urban land space planning in the new period, and further develop the concept of ecological civilization construction and sustainable development.

## 2. Materials and Methods

### 2.1. Study Area and Data Preprocessing

Wuhan City (Figure 1) is located in the central region of China, located in the east of Hubei Province, between 29°58′–31°22′ north latitude, and 113°41′–115°05′ east longitude. Wuhan City is the capital of Hubei Province, the only sub-provincial city of six central provinces, the central city of central China, and the core city of the Yangtze River Economic Belt. Located in the eastern part of Jianghan Plain, Wuhan enjoys unique and thick water owing to its ecological advantages. It has numerous rivers and lakes in the city, and is well equipped with natural resources such as mountains, rivers, forests, fields, and lakes. There are 446 mountains, and the water area accounts for approximately a quarter of the total area of the city [20,44,45].

Since the reform and opening up, the development of Wuhan’s territorial space has become increasingly active. During the accelerated development of urban industrialization and urbanization, the quantitative structure and spatial layout of Wuhan’s territorial space have undergone great changes. Although the development of territorial space has supported the sustainable and rapid growth of the urban economy and society, it has caused numerous problems such as resource constraint tightening, environmental condition deterioration, and low land-use efficiency, which have posed a certain threat to the high-quality and sustainable development of the city [46,47].

According to the “Wuhan Social and Economic Statistics Bulletin”, by 2020, the city’s permanent resident population of 10.8929 million, the city’s GDP 1561.06 billion CNY, and the urbanization rate of 80.04% [48]. In the same period, the average annual temperature ranged from 15.8–17.5 °C, and the average annual precipitation was 1269 mm. Therefore, we must strictly control urban land expansion and delimit urban growth. In addition, the population of the main urban areas should be limited to a maximum of 5.02 million to avoid excessive population agglomeration [47,48,49]. A new city was designed and built in the six remote urban areas, forming a “main city + new city group” and a multi-axis and multi-center urban overall spatial framework with the main urban area as the core. Wuhan is an important industrial, scientific, and educational base in China, and is also a comprehensive transportation hub.

The data used in this study are mainly divided into two parts: natural environment element data and social economic data. Land-use data are the high-resolution satellite images of Wuhan 2020 downloaded by the author based on the current land-use data of Wuhan 2020 from the geographic monitoring cloud platform (http://www.dsac.cn/, accessed on 17 November 2021) and the Google map data in the “Loca Space Viewer V1.0” software. They were obtained by visual interpretation according to the classification standard of investigation work in the technical specification of “three Tones”. Other data information and their sources are listed in Table 1.

### 2.2. Methodology

#### 2.2.1. Modeling Framework

The nondominated sorting genetic algorithm (NSGA) is a multi-objective genetic algorithm based on the Pareto frontier proposed by Srinivas and Deb in 1994. Based on NSGA, the improved second-generation (NSGA II) and third-generation (NSGA II) algorithms of the NSGA series were proposed in 2002 and 2014, respectively. The core concept of these algorithms is to select the optimal individual by non-dominated ranking and controlling population diversity. NSGA II is the most popular and widely used algorithm in the series (20,023 citations on Web of Science as of 1 September 2020), while NSGA II is the most recently proposed and state-of-the-art algorithm. The NSGA II algorithm has been widely used in the field of land-use optimization. To optimize the three types of spatial layout in Wuhan, this study first proposed the optimization of the land-use layout, and finally obtained the optimization results of the three types of spatial layout by constructing three types of spatial classification systems. The main research steps are as follows. (1) To maximize the eco-economic benefits in urban development and address the constraints determined by specific policies and planning documents, a quantitative structure optimization model of land use was constructed. (2) The solution sets under the three conditions of maximum ecological benefit, maximum economic benefit, and maximum comprehensive benefit in the Pareto optimal solution obtained by NSGA II were selected as the three scenarios of urban land-use optimization. (3) Based on the quantitative optimization results, the simulation results of land-use spatial layout were obtained using the FLUS model. (4) The spatial layout optimization results of urban space, ecological space, and agricultural space in the target year were obtained by combining the cohesion system of “three types” of space and land-use types.

#### 2.2.2. Land-Use Demand Structure Optimization

We chose NSGA II to solve the MOP problem, as it can maintain the diversity of solutions, have strong global optimization ability, improve the robustness of the algorithm, and avoid the algorithm falling into the local optimal. The steps are briefly summarized as follows: (1) We transformed the remote sensing image of land use in 2020 into a two-dimensional matrix, and input the data into Matlab to generate the initial species group. (2) We used Matlab to calculate the suitability value of the three objective functions. Fast non-dominant sequencing and crowded degree calculation were carried out, and then the suitable parent generation was selected through competitive bidding, and the progeny population was obtained through crossover (simple single-point crossover, multi-point crossover) and mutation (polynomial variation). (3) Fast non-dominated sorting and crowding degree calculation were carried out for children and parents, and the first N = 60 solutions were selected to determine whether the number of iterations was reached. (4) If the number of iterations was not reached, we returned to the second step. (5) Upon reaching the convergence condition, the most suitable individuals from Pareto solution set were selected, and became the solution to optimize the quantity structure of land use. The MOP model generally consists of a decision variable parameter, an objective function, and a constraint condition.

#### 2.2.3. Objective Function

According to the types and characteristics of land and resources utilization in Wuhan City, this study comprehensively determined the decision variables in the MOP model [47,48,49]. A total of nine decision variables including cultivated land, garden land, rural construction land, woodland, grassland, water area, unused land, wetland, and urban construction land were set. Based on the basic land-use principle of “combining economic benefits, ecological benefits and social benefits”, the objective function was constructed by comprehensively considering the requirements of Wuhan’s economic development and ecological environment protection. The social benefit objective is mainly reflected in the specific requirements of macro policy documents for various types of land [13]; therefore, it runs through the construction process of constraint conditions. The objective function is expressed as follows.
(1)$$f_{n}\left(x\right)=\sum_{i=1}^{n}w_{i}x_{i}$$ where $f_{n}\left(x\right)\;$ is the value of economic or ecological benefits; $w_{i}\;$ is the utilization efficiency coefficient of quasi-land-use type; $x_{i}\;$ is the area of quasi-land-use type.

Ecological benefit target

Based on the actual value of average grain yield per unit area in Wuhan from 2000 to 2020 [47], this study predicted that the actual value of various grain per unit area in 2025 will be 2.3 × 10^6^ CNY/km^2^. The value of a single equivalent factor is approximately 1/7 of the actual value of average grain yield per unit area, and the single equivalent factor in Wuhan’s ecosystem services in 2025 was calculated to be 3.2×10^5^ CNY/km^2^. According to the value equivalent factor table [47], the ecosystem service value per unit area of each land-use type in Wuhan was calculated (Table 2).

2.Economic benefit target

In this study, the ratio of the output value corresponding to the land-use type and its area was regarded as the economic efficiency coefficient of all kinds of land use. Based on the output benefits per unit area of various types of land use in Wuhan in 2000, 2005, 2010, 2015, and 2020 and the areas of various types of land use in the corresponding years, the corresponding economic value coefficients of Wuhan in the above-mentioned five years were calculated. The grey prediction GM (1,1) model was used to further predict the economic value coefficients of all types of land in Wuhan in 2025 (Table 2).

#### 2.2.4. Constraints

1.***Constraints on total national space.*** Wuhan’s administrative area was selected for this study. Before and after optimization, the total research area should be consistent and equal to the total area of Wuhan’s administrative area, as follows:


(2)
$$\sum_{i=1}^{9}x_{i}=8600\;{\mathrm{km}}^{2}$$


2.***Constraints on cultivated land.*** By the end of the 14th Five-Year Plan of Wuhan Municipal Land Space (hereinafter referred to as the Plan), Wuhan will have built 2.47 million mu of high-standard farmland. According to the requirements of the Land Management Law, the permanent basic farmland should generally account for more than 80% of the cultivated land in the administrative area:


(3)
$$x_{1}\times 80\%\geq {\;1650\;\mathrm{km}}^{2}$$


3.***Constraints on rural construction land.*** The area of rural residential land will be reduced following the acceleration of the land consolidation process and the requirements of the new rural construction. However, considering the regular needs of rural residents, this study stipulated that the area of rural construction land should not be lower than the current area of residential land:


(4)
$$310\;{\mathrm{km}}^{2}\leq x_{3}\leq {443\;\mathrm{km}}^{2}$$


4.***Woodland constraint.*** As an important part of the ecosystem, woodland has important functions such as ecological protection, and soil and water conservation. The “plan” requires the adoption of artificial afforestation, closing mountains for forest cultivation and other ways of strengthening the restoration of mountain vegetation. By 2025, the amount of forest land in Wuhan should reach 1,777.8 km^2^:


(5)
$$x_{4}\geq 1780\;{\mathrm{km}}^{2}$$


5.***Water area constraint.*** Wuhan covers approximately a quarter of the total water area under its jurisdiction, and is known as the city of 100 lakes. The Master Plan for Wuhan’s Territorial Space (2021–2035) calls for the establishment of the most stringent water resource management system in the new development period to ensure that water areas are not damaged. Furthermore, considering that the complexity of transforming part of the construction land into water area, the predicted value calculated by the annual average growth rate in the recent ten years was regarded as the upper limit, and then the constraint condition of the water area was set as follows:


(6)
$${2100\;\mathrm{km}}^{2}\leq x_{6}\leq {2395\;\mathrm{km}}^{2}$$


#### 2.2.5. FLUS Model

The selection of reasonable driving factors can effectively improve the precision of the model. This study, according to the characteristics of the distribution of land and resources in Wuhan city, starting from the availability of data predicting accuracy, chose the elevation, slope, and the five natural elements including the population and GDP data and three social economic data as driving factors and set the resolution of the data unity to 30 m. 

In the training of random forest RF algorithm, all data on the driving factors were input first, and the sampling ratio was 20‰ by uniform distribution. Finally, the probability of suitability of all types of land was obtained, which reflected the interaction and competition between land types. In the FLUS model, the probability of land conversion depends on the probability of driving factors and the density of the neighborhood output of the neural network. Moreover, it is affected by the inertia coefficient and conversion cost. The inertia coefficient of the first type of land in iteration time is expressed as follows:$$Inertia_{k}^{t}=\{\begin{array}{l} Inertia_{k}^{t-1}{}_{}\;\mathrm{if}\;\left|D_{k}^{t-2}\right|\leq \left|D_{k}^{t-1}\right|\\ Inertia_{k_{}}^{t-1}\times \frac{D_{k}^{t-2}}{D_{k}^{t-1}}\;\mathrm{if}\;0>D_{k}^{t-2}>D_{k}^{t-1}\;\\ Inertia_{k}^{t-1}\times \frac{D_{k}^{t-2}}{D_{k}^{t-1}}{\;\mathrm{if}\;D}_{k}^{t-1}>D_{k}^{t-2}>0 \end{array}$$
$$TProb_{p,k}^{t}=sp\left(p,k,t\right)\times \Omega_{p,t}^{t}\times Intertia_{k}^{t}\times \left(1-sc_{c\to k}\right)$$ where $TProb_{p,k}^{t}\;$ is the cost of changing land-use type to land-use type; $\Omega_{p,t}^{t}\;$ is the degree of difficulty that conversion occurs; $Intertia_{k}^{t}$ acts as a neighborhood. The neighborhood influence factor is used to reflect the interaction between different land-use types and different land-use units within the neighborhood [20]. In this study, the 3 × 3 Moore field was adopted, and the number of iterations was set to 300. In order to verify the simulation accuracy of FLUS model, the 2015 data of Wuhan city was taken as the base period to simulate the spatial distribution in 2020. Comparing the result with the actual situation in 2020, the overall accuracy simulated by the FLUS model was 95.28%, and the Kappa coefficient was 0.893, showing an excellent simulation effect of spatial distribution. The model and its parameters are therefore suitable as the basis of this study.

#### 2.2.6. Spatial Classification System

The Outline of National Land Planning (2016–2030) divides national land space into three categories: urban space, agricultural space, and ecological space. However, there are only a few studies on the classification of the three categories, and no unified classification standard has been formed yet. Based on the functions of urban, agricultural, and ecological space [22,23,24,25,26], this study combined the existing research results and the actual situation of the research area, connected the three types of space with the work classification standard of “three tones”, and divided the three types of space (Table 3).

## 3. Results

### 3.1. Land Use of Multi-Objective Optimization

NSGA II was used to establish the quantitative optimization model constructed above. Studies have shown that the genetic algorithm is random and the results of each run are different. However, the rule of one test is consistent with that of multiple tests [19]. After several experiments, the population size of the algorithm in this study was set to 40. Crossover probability was set to 0.8, mutation probability to 0.1, constraint tolerance to 0.001, and the number of iterations to 1000.

In this study, based on the 67 Pareto optimal solutions obtained by NSGA II, three solutions with the largest sum of economic benefits, the largest sum of ecological benefits, and the largest total comprehensive ecological and economic benefits were selected as the results of quantitative optimization (Figure 2 and Table 4). The results of land-use quantity optimization in Wuhan City under economic priority, ecological priority, and comprehensive consideration of economic and ecological benefits are presented, respectively. In addition, the simulation results of the spatial layout were obtained based on the FLUS model. Finally, according to the three types of spatial classification system constructed, the land-use optimization results were transformed into three types of spatial optimization results, and the land space layout optimization method of Wuhan was obtained.

### 3.2. Quantitative Structure Optimization

The reduction in the requirement for permanent prime farmland protection was transferred to the other two types of space. Among them, under the ecological benefit priority scenario, the agricultural space area decreased by approximately 54.4%, and the optimized area only accounted for 29.19% of Wuhan’s land area, which indicated that the contribution of cultivated land, garden land, and rural construction land to the urban ecological service value was low.

Under the ecological protection priority scenario, the ecological space area of Wuhan increased by 1002.43 km^2^, and the area proportion reached the maximum of the three scenarios. Particularly, the area of forest land increased significantly, and the area of wetland and water area increased slightly. Under the other two scenarios, the area of ecological space increased, reflecting the city’s requirements for strengthening mountain vegetation restoration and other related ecological restoration construction, and providing development space for Wuhan to establish a national demonstration city of ecological civilization construction.

During the new development of Wuhan into a national central city and a core city in the Yangtze River Economic Belt, its economy will continue to grow rapidly, and the demand for urban construction land will not be reduced. Under the three development scenarios, the area of urban space exhibited a relatively large increase. In the scenario of optimal comprehensive benefit and priority of economic benefit, the urban space increased by 54.67%, while in the scenario of ecological priority, owing to the strict protection of ecological land and emphasis on ecological service value in the system area, the area increase of urban space was lower than that of the other two cases, only increasing by 359.74 km^2^.

### 3.3. Spatial Layout Simulation

The FLUS model was used to analyze the Wuhan national spatial layout optimization; in the economic benefit of priority situation, for the agricultural space, the central city almost no longer had arable land, and reduced and the Huangpi District in the south of the Jiangxia District’s northwest area had more. While most agricultural space transformed into ecological space, there was also a part of the agricultural space which transformed into urban space. Under the scenario of optimal comprehensive benefits, the ecological space in the southwest and northwest areas of Wuhan was partially transformed into agricultural space, which provides favorable conditions for large-scale agricultural management. Under the ecological benefit priority scenario, part of the agricultural space in the northeast of Wuhan was transformed into ecological space, and part of the cultivated land patches had to be divided, resulting in a large degree of landscape fragmentation.

In addition, the ecological space of Wuhan in the base year was dominated by Huangpi District in the northwest, followed by the east side of Jiangxia District, where a large amount of woodland and grassland grew. In 2025, after the ecological benefit as the first optimization simulation, ecological space expansion of the large area, Jiangxia District, Wuhan was the most significant, while Caidian District and Xinzhou District’s ecological space areas also increased obviously, and even more for the transformation of the agricultural space. Mainly in the three areas, to drive the sustainable development of Wuhan city’s ecological protection, we should realize the farmland’s restoration into forest to implement the requirements of vegetation restoration. However, under the scenario of comprehensive benefits, the area change of the southeast area of Wuhan was relatively obvious. This area is mainly covered by Jiangxia District of Wuhan, including Liangzi Lake area. Agricultural space is obviously transformed into ecological space, and Liangzi Lake was the core area for expansion.

As shown in Figure 3, the urban space mainly expanded from the central urban area to the surrounding areas in the “spreading the pie” manner, and in the process, mainly encroached on agricultural space such as farmland and garden land, and occupied part of the ecological space, including woodland. Compared with economic priority and the comprehensive optimal scene, the ecological priority scenario northwest of downtown urban space occupied relatively little agricultural space, thereby indicating its limitation by the administrative area in Wuhan city boundary. The urban space mainly expanded in the north–south direction, and prioritized the northwest, which could drive the construction and development of the airport economic zone in the city of Wuhan.

## 4. Discussion

Against the backdrop of advanced construction of the ecological civilization era, the expansion of the urban space will inevitably be affected by a certain degree of restraint. However, during the rapid development of big cities such as Wuhan, to constantly improve the population and its development needs, urban spatial expansion is inevitable; therefore, improving the efficiency of three kinds of spatial land use is the main objective of future national spatial planning. Theodor used the genetic algorithm and applied it to a specific land-use planning problem in The Netherlands, with a good performance in dealing MOP problems [47,50]. The NSGA II algorithm adopts the elite retention strategy to speed up the convergence speed of the algorithm, and uses the fast non-dominated sorting and crowding operator comparison strategy, which improves the diversity of heavy groups and prevents the algorithm from falling into local optimum. The convergence of the algorithm should be sped up. The NSGA II algorithm based on the Pareto optimal solution has more optimized performance in dealing with multi-objective problems. In this study, economic benefits and ecological benefits were set as objective functions. According to the characteristics of urban development in Wuhan, the ecological benefits considered the ecological service value as the parameter, and the NSGA II algorithm could be used for a more reasonable and scientific optimization of the future land-use quantity structure. The model performed spatial optimization according to the results of quantitative structure optimization, natural conditions, and economic factors. Under the three optimization results, the GDP exceeded that preceding optimization, which could better meet the needs of sustainable ecological development and provide a sustainable development plan for land use in Wuhan. Compared with some other studies [24,50,51], this paper constructed the ecosystem service value coefficient of the target year and also considered the economy and ecology so that the objective function obtained was more accurate. The spatial pattern of future land use obtained was closer to the real value by adding the influence of future development planning policies into the land-use simulation.

Based on our research results and the simulated land-use spatial layout of Wuhan under the scenario of economic and ecological priority in the future as shown in Figure 3, the following three suggestions are put forward for the optimization of land space in Wuhan. The following recommendations are proposed for the land space optimization:1.Considering the decreasing agricultural space, the contiguous-area arable land with high efficiency should be designated as the permanent primary farmland-centralized protection zone, focusing on the development of large-scale agricultural operation and production modes, improving the efficiency of grain production per unit area, and ensuring that the local food demand is met within the effective agricultural production space. In addition, the construction of ecological agricultural areas should be studied comprehensively considering the economic and social benefits of agricultural space, focusing on digging ecological benefits, and realizing the complementarity between ecological space and agricultural space.2.It is recommended to rely on the rich ecological resources of Wuhan, including land and water corridors and other resource elements, to build a natural ecological network integrating mountains, rivers, forests, fields, and lakes, and expand the green open space. To explore the social value and economic benefits of ecological space, certain fragmented waters should be integrated with residential areas, spatial control systems should be established based on the features of different ecological spaces, differentiated control requirements need to be proposed, a shift from rigid protection to equal emphasis on control and guidance is required, and conflicts with agricultural production and urban construction must be avoided.3.The efficiency of urban space development and utilization should be improved, and differentiated renewal and renovation policies established. Inefficient and idle land in cities, from the bottom of the sewers, should be utilized, and regulations put in place to strengthen the clean-up of batch and unused land, using stock planning as the main line, tearing open change to gradually combine the old city renovation projects, further enhance the development of the urban space utilization, reduce regional ecological security, improve food security, and reduce the conflict between economic development and construction. Based on the distribution pattern of natural resources, the urban development and construction pattern of “multi-center, network, and group” should be built according to the group structure.4.In terms of land-use layout optimization, under the ecological protection scenario, the layout of central towns was more regular, the overall degree of fragmentation was moderate, and the overall layout of land-use changed from centralized development to balanced development. The degree of landscape fragmentation was improved, and the overall layout of land use had the trend of “urban space > ecological space”. Under the ecological benefit priority scenario, the layout of central towns was more regular, the overall degree of fragmentation was improved, the overall ecological benefits were significantly improved, and the overall layout of land use had the trend of “ecological space > urban space”. In the process of urbanization in Caidian District, attention should be paid to sustainable land use, ecological protection should be strengthened on the basis of ensuring economic benefits, and land structure and layout optimization should be promoted through the integration of natural resources and the comprehensive improvement of land.

In this study, the objective function of ecological benefits was constructed using the relatively basic equivalent factor method, and some social service values were disregarded when estimating the economic value of wetlands. In addition, because we were limited by data acquisition, this study only built constraints based on relevant documents of the 14th Five-Year Plan of Wuhan. In view of the abovementioned problems, future research could use the InVEST model, energy analysis, and other more scientific methods of urban ecological asset accounting and quantification to determine the ecological benefits of various types of land use. Furthermore, the quantitative structure of urban spatial resources in future can be predicted comprehensively using a mathematical model and macro-control to obtain more rigorous research results.

## 5. Conclusions

To promote the high-quality development of national spatial layout and the support system in the future, this study focused on the uniqueness of the Wuhan city planning MOP model, and used the genetic algorithm to yield Pareto frontier science solutions and select the ecological and economic benefit of preference as well as the optimal comprehensive benefits of three scenarios of the optimal solution set. Furthermore, three types of spatial layout simulation results were obtained using the FLUS model. The analysis results showed that:

1.The joint use of the MOP model based on NSGA II solution and FLUS model to combine nature, traffic, and social and economic characteristics could effectively solve problems related to land-use structure and spatial layout optimization. Furthermore, the national spatial pattern was further optimized using the three types of space classification systems. From the perspective of space in the future urban resource configuration mode, this is an innovative analysis method.2.In terms of optimal allocation of quantity, under the three development scenarios, the agricultural space showed a significant decline, while the ecological space and urban space increased. The decline rate of agricultural space was ordered as follows: ecological priority scenario > optimal comprehensive benefit scenario > economic priority scenario, which basically only meets the protection requirements of permanent basic farmland with respect to quantity, allowing for ecological and urban development.3.In terms of spatial layout simulation, the urban space mainly expanded longitudinally in the north–south direction in a “spreading the pie” manner. In the ecological space scenario, part of the cultivated land in the northeast of the city was occupied owing to the considerable increase in ecological space, leading to a high degree of landscape fragmentation, which is not conducive to large-scale agricultural management. However, under the optimal comprehensive benefit, part of the fragmented ecological space in the western part of Wuhan was transformed to agricultural space, which provides favorable conditions for agricultural large-scale operation.4.In the MOP–FLUS model coupling, due to the variability of government policy planning documents, the setting of model constraints was not comprehensive enough, which may cause deviations from the actual development situation. How to more comprehensively consider factor structure in the model should be investigated to make the analysis results more accurate, and this be the focus of future research on the optimal layout and allocation of land space.

## Figures and Tables

**Figure 1 ijerph-20-00495-f001:**
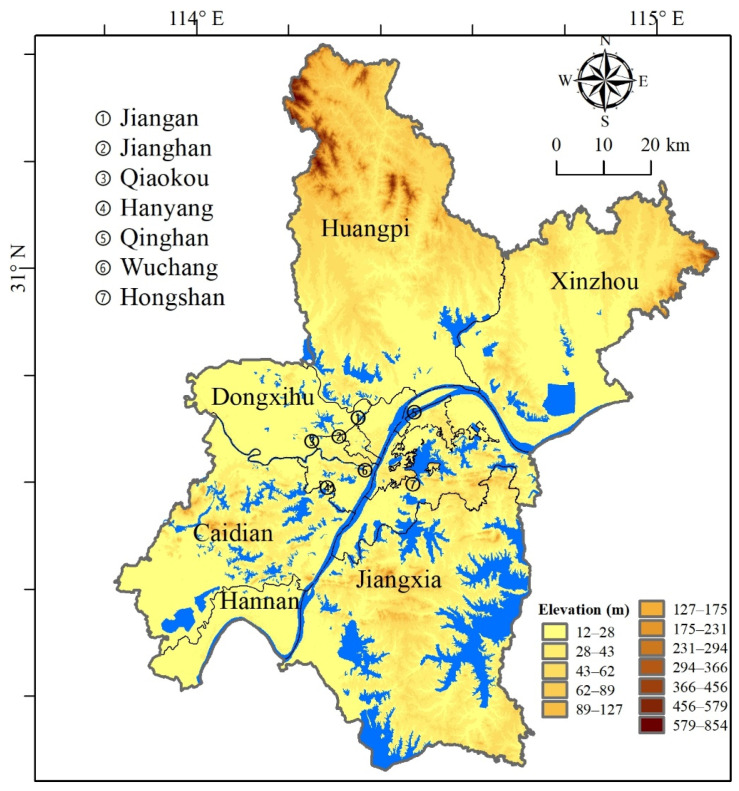
Location of the Wuhan Urban Agglomeration Area.

**Figure 2 ijerph-20-00495-f002:**
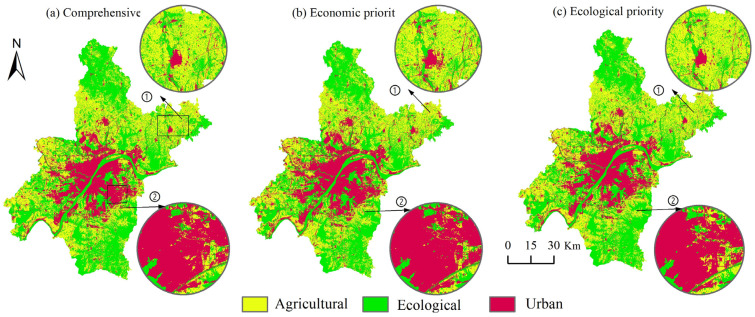
Three kinds of spatial layout optimization results.

**Figure 3 ijerph-20-00495-f003:**
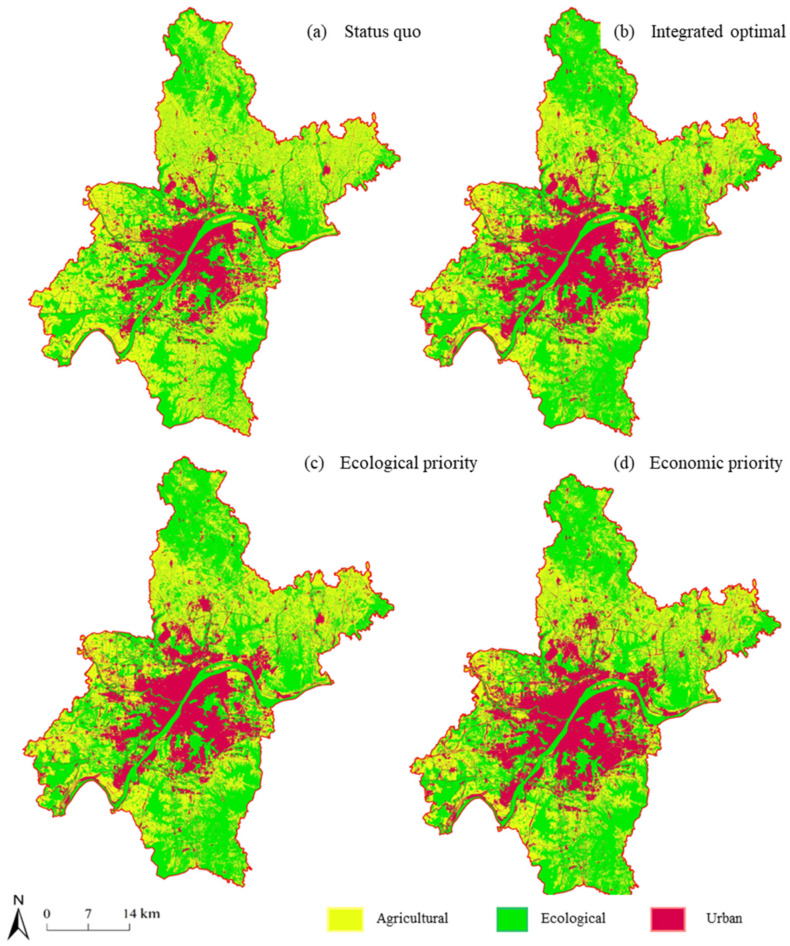
Partial details of optimization results of three types of spatial layout.

**Table 1 ijerph-20-00495-t001:** Research data and their sources.

Data Type	Data	Data Source	Description
Natural environment data	30 m resolution DEM data in 2021	Geospatial data cloud	Used to calculate the probability of suitability
	Soil data at 1 km resolution for 2008	Harmonized World Soil Database v 1.2	Used to calculate the probability of suitability
	Average temperature at 1 km resolution in 2015	WorldClim version 2.0	Used to calculate the probability of suitability
Social and economic data	Population density data at 100 m resolution in 2015	Worldpop	Used to calculate the probability of suitability
	Road network data in 2020	OpenStreetMap	Used to calculate the probability of suitability
	2015 1 km resolution GDP dataset	China’s GDP spatial distribution kilometer grid dataset	Used to calculate the probability of suitability
	Classified total output value of agriculture, forestry, animal husbandry and fishery in 2020	Wuhan Bureau of Statistics	Objective function determination

**Table 2 ijerph-20-00495-t002:** Coefficient of the objective function.

Coefficient	Cropland	Garden	Rural Settlement	Woodland	Grassland	Waters	Unused Land	Wetland	Urban Construction
$f_{1}$	13	40	0.7	66	37	409	4	170	0.7
$f_{2}$	13	208	140	12	191	70	0	0	12,009

Unit: 10^5^ CNY/km^2^.

**Table 3 ijerph-20-00495-t003:** Classification system of three types of land.

Classifications of “Production–Living–Ecological”	Land Use
Agricultural land	Cropland
	Garden land
	Rural settlement
Ecological land	Woodland
	Grassland
	Waters
	Wetland
	Unused land
Urban land use	Urban construction

**Table 4 ijerph-20-00495-t004:** Area statistics before and after “three categories” space optimization.

Three Types	Area	Integrated Optimal	Ecological Priority	Economic Priority
Agricultural land	3866	2601	2504	2706
Ecological land	3853	4648	4855	4544
Urban land use	858	1328	1218	1328

## Data Availability

Not applicable.

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
