# Peer review of "Compound Optimization of Territorial Spatial Structure and Layout at the City Scale from “Production–Living–Ecological” Perspectives"

_ijerph, 2022, doi:10.3390/ijerph20010495_

Round 1
Reviewer 1 Report
Comments on [IJERPH] Manuscript ID: ijerph-2059693
Based on the MOP (Multi-Objective Programming) and FLUS (Future Land Use Simulation) model, this paper examined 14 composite optimization of the spatial layout of Wuhan city. Overall, the paper is structured in a solid way, with a detailed description of the research context, methodologies, and results. As a reviewer, however, my problem with this paper rests in the clarity and validity of the methodologies and terms used.
1. What scientific problem is addressed in this paper? What is its contribution?
2. In the section of "2.1. Study Area and Data Preprocessing", the author spends a lot of space describing the location, natural environment and economic development of Wuhan, but these contents are too scattered. It is not obvious that these words are very closely related with "compound optimization of territorial spatial structure and layout at city scale from spatial perspectives".
3. In "Table 1. Research data and its sources", some data are no year marked, some are 2008, some are 2015, and some are 2020. Why is it not a uniform data, and what is the reason?
4. The paper use the MOP model and FLUS model,ok, the accuracy of the model prediction needs to be verified, maybe a small area can be selected for comparative verification.
5.Why does the area of cultivated land in these scenarios show the following characteristics: ecological priority scenario > optimal comprehensive benefit scenario > economic priority scenario?Could you explain it more clearly and thoroughly?
6. Why does the city expand from north to south? What is the internal mechanism? Does the expansion differ in different regions?
7. The "conclusion" of the article does not give the description of regularity, it lacks the specific description of the contribution to relevant scientific models.
8.This paper aims to analyze the Compound optimization of territorial spatial structure and layout at city scale from spatial perspectives, It is suggested that the authors separately focus on compound optimization mode or path of land space structure of Wuhan City in 2025 under different scenarios.
9.In addition, the format of the references needs to be uniform.
Author Response
REVIEWER #1
Comment #1: What scientific problem is addressed in this paper? What is its contribution?
Response #1: We appreciate the expert's responsible advice. In this manuscript, we constructed a conductive framework to exploit ecosystem and economic benefits in spatial layout optimization. Land use pattern optimization can significantly offset ecological deterioration caused by urban sprawl, yet there is no specialized method to embody this merit. We coupled MOP (Multi-Objective Programming) with FLUS (Future Land-Use Simulation) model to simulate two ecological scenarios.
Comment #2: In the section of "2.1. Study Area and Data Preprocessing", the author spends a lot of space describing the location, natural environment and economic development of Wuhan, but these contents are too scattered. It is not obvious that these words are very closely related with "compound optimization of territorial spatial structure and layout at city scale from spatial perspectives".
Response #2: We appreciate the expert's responsible advice. We have revised and
adjusted the section of "2.1. Study Area and Data Preprocessing" of the paper. We have revised the paper as follows:
“According to the "Wuhan Social and Economic Statistics Bulletin", by 2020, the city's permanent resident population of 10.8929 million, the city's GDP 1561.06 billion CNY, the urbanization rate of 80.04%. In the same period, the average annual temperature ranges from 15.8–17.5°C, and the average annual precipitation is 1269 mm. In addition, the population of the main urban areas should be limited to a maxi-mum of 5.02 million to avoid excessive population agglomeration. A new city was de-signed and built in the six remote urban areas, forming a "main city + new city group" and a multi-axis and multi-center urban overall spatial framework with the main urban area as the core. Wuhan is an important industrial, scientific and educational base in China. It is also a comprehensive transportation hub, known as the "thoroughfare of Nine Provinces".
Comment #3: In "Table 1. Research data and its sources", some data are no year marked, some are 2008, some are 2015, and some are 2020. Why is it not a uniform data, and what is the reason?
Response #3: We appreciate the expert's responsible advice. We added the explanation of the lack of years. The reason for the inconsistency of data years is mainly because it is difficult to unify all data into one year in the process of data collection. We tried our best to use the latest data, and if we could not find suitable data, we would use other years to replace it.
Comment #4: The paper use the MOP model and FLUS model, ok, the accuracy of the model prediction needs to be verified, maybe a small area can be selected for comparative verification.
Response #4: We appreciate the expert's responsible advice. We have revised the paper as follows:
“In order to verify the simulation accuracy of FLUS model, the 2015 data of Wuhan city was taken as the base period to simulate the spatial distribution in 2020. Comparing the result with the actual situation in 2020, the overall accuracy simulated by FLUS model was 95.28%, and the Kappa coefficient was 0.893, showing excellent simulation effect of spatial distribution. The model and its parameters are suitable as the basis of this study.”
Comment #5: Why does the area of cultivated land in these scenarios show the following characteristics: ecological priority scenario > optimal comprehensive benefit scenario > economic priority scenario?Could you explain it more clearly and thoroughly?
Response #5: We appreciate the expert's responsible advice. I am very grateful for your interest in these points. According to Table 3, the agricultural space includes cultivated land, garden land and rural residential areas. According to the correlation coefficient in Table 2, it can be concluded that the coefficient of our cultivated land is the same in the two scenarios, but different garden land and rural residential areas. So not arable land, but agricultural space ecological priority scenario > optimal comprehensive benefit scenario > economic priority scenario.
Comment #6: Why does the city expand from north to south? What is the internal mechanism? Does the expansion differ in different regions?
Response #6: We appreciate the expert's responsible advice. Traffic/Road network, elevation (DEM) and economic development level are the main driving factors of urban land use change in Wuhan. We used the same driving factors to analyze the adaptive development probability of the whole land use type in Wuhan.
Comment #7: The "conclusion" of the article does not give the description of regularity, it lacks the specific description of the contribution to relevant scientific models.
Response #7: We appreciate the expert's responsible advice. We have revised the paper in the "conclusion" as follows:
In the MOP-FLUS model coupling, due to the variability of government policy planning documents, the setting of model constraints is not comprehensive enough, which may deviate from the actual development situation. How to consider more comprehensively in the model factor structure to make the analysis results It will be more accurate and will be the focus of research on the optimal layout and allocation of land space in the future.
Comment #8: This paper aims to analyze the Compound optimization of territorial spatial structure and layout at city scale from spatial perspectives, It is suggested that the authors separately focus on compound optimization mode or path of land space structure of Wuhan City in 2025 under different scenarios.
Response #8: We appreciate the expert's responsible advice. We have revised the paper in the “Discussion” as follows:
In terms of land use layout optimization, under the ecological protection scenario, the layout of central towns is more regular, the overall degree of fragmentation is moderate, and the overall layout of land use changes from centralized development to balanced development; The degree of landscape fragmentation is improved, and the overall layout of land use is in the trend of "urban space > ecological space". Under the ecological benefit priority scenario, the layout of central towns is more regular, the overall degree of fragmentation is improved, the overall ecological benefits are significantly improved, and the overall layout of land use is The situation of "ecological space>urban space". In the process of urbanization in Caidian District, attention should be paid to sustainable land use, ecological protection should be strengthened on the basis of ensuring economic benefits, and land structure and layout optimization should be promoted through integration of natural resources and comprehensive improvement of land.
Comment #9: In addition, the format of the references needs to be uniform.
Response #9: We appreciate the expert's responsible advice. I am very grateful for your interest in these points. We checked the format of the references in whole, matching the journal's requirements
Reviewer 2 Report
1) There is a certain difference between the abstract and the article text itself. The abstract has a majot focus on the urban sprawl and the formal development of the urban territory meanwhile the text is concentrated on the use of technology in urban mapping.
2) The negative impacts of the urban growth are mentioned in the abstract but not clearly explained in the text.
3) What does the "spreading the pie" manner mean in the case of Wuhan?
4) How do the results permit, stimulate or enhance the sustainable development in Wuhan? How do the national public policy define the mentioned "national space layout and support system for high-quality development"?
5) Could there be a little bit more of explanation of the concepts of urban pattern of "two rivers and three towns" and"the thoroughfare of nine provinces"?
6) A diagram would help to understand better the concepts of "multi-axis and multi-center, one main city and six new towns, and numerous towns".
7) Could there be a map or diagram in order to understand: "Wuhan is limited to the Third Ring Road, and the plan is to build a satellite city in each of the six remote urban areas, thereby creating a "1+6" urban basic pattern"?
8) In general, maps and diagrams would be needed to explain Wuhan's current situation in the Chinese context, especially considering readers from other regions of the world. Also, in order to make clear the results and findings of the research work done, more images and maps would be needed. The captions should explain better the contents of the images.
9) Finally, what specific benefits do the results of the research work done provide to he future planning of the Wuhan region and the public policy?
Author Response
REVIEWER #2
Comment #1: There is a certain difference between the abstract and the article text itself. The abstract has a majot focus on the urban sprawl and the formal development of the urban territory meanwhile the text is concentrated on the use of technology in urban mapping.
Response #1: We appreciate the expert's responsible advice. I am very grateful for your interest in these points. We have revised the paper as follows:
Land-use optimization, as an important resource allocation method, can be defined as the process of allocating various activities to different geographic units. How to manage and control land expansion has become an urgent issue, leading a series of problems such as environmental damage and a sharp decrease in cultivated land, leading to unfavorable phenomena such as excessive urban expansion, occupation of cultivated land and important ecological spaces, and overheating of real estate development. Based on the land use data of Wuhan city in 2020, a coupling MOP (Mul-ti-Objective Programming) and FLUS (Future Land Use Simulation) model was used to examine the national spatial structure and the optimization of the spatial layout. Our results show that: (1) In terms of quantitative optimal allocation, the ecological space, and urban space increased, while the agricultural space greatly decreased under the three development scenarios. (2) In the simu-lation of spatial layout, the urban space mainly expands vertically in the north-south direction. In the ecological space scenario, the ecological space occupies part of the cultivated land in the northeast of the city, resulting in a high degree of landscape fragmentation, which is not conducive to large-scale agricultural management. However, under optimal comprehensive benefit, part of the fragmented ecological space in the western part of Wuhan is transformed into an agricultural space. (3) A combination of the MOP and FLUS models can effectively determine land use structure and address spatial layout optimization problems and can project space in the future urban re-source configuration mode. This finding can provide a reference for the optimization of the spatial structure and layout of similar cities.
Comment #2: The negative impacts of the urban growth are mentioned in the abstract but not clearly explained in the text.
Response #2: We appreciate the expert's responsible advice. I am very grateful for your interest in these points. We have removed related unclear descriptions.
Comment #3: What does the "spreading the pie" manner mean in the case of Wuhan?
Response #3: We appreciate the expert's responsible advice. I am very grateful for your interest in these points. We have removed related unclear descriptions.
Comment #4: How do the results permit, stimulate or enhance the sustainable development in Wuhan? How do the national public policy define the mentioned "national space layout and support system for high-quality development"?
Response #4: We appreciate the expert's responsible advice. I am very grateful for your interest in these points. We have added related descriptions in “Discussion” and “Conclusion”.
Comment #5: Could there be a little bit more of explanation of the concepts of urban pattern of "two rivers and three towns" and "the thoroughfare of nine provinces"?
Response #5: We appreciate the expert's responsible advice. I am very grateful for your interest in these points. We have removed related unclear descriptions.
Comment #6: A diagram would help to understand better the concepts of "multi-axis and multi-center, one main city and six new towns, and numerous towns".
Response #6: We appreciate the expert's responsible advice. I am very grateful for your interest in these points. We have removed related unclear descriptions.
Comment #7: Could there be a map or diagram in order to understand: "Wuhan is limited to the Third Ring Road, and the plan is to build a satellite city in each of the six remote urban areas, thereby creating a "1+6" urban basic pattern"?
Response #7: We appreciate the expert's responsible advice. I am very grateful for your interest in these points. We have removed related unclear descriptions.
Comment #8: In general, maps and diagrams would be needed to explain Wuhan's current situation in the Chinese context, especially considering readers from other regions of the world. Also, in order to make clear the results and findings of the research work done, more images and maps would be needed. The captions should explain better the contents of the images.
Response #8: We appreciate the expert's responsible advice. We have clearly shown the necessary maps in the paper.
Comment #9: Finally, what specific benefits do the results of the research work done provide to the future planning of the Wuhan region and the public policy?
Response #9: We appreciate the expert's responsible advice. I am very grateful for your interest in these points. We have added related descriptions in “Discussion” and “Conclusion”
Reviewer 3 Report
I would like to see some complementary images of land uses in the case study, for the international readers that are not from China.
I suggest to add in the conclusions how this spatial methods can be integrated in a national land policy, especially related to environmental conservation.
How this spatial analysis and tools can be socialized into community?
Author Response
REVIEWER #3
Comment #1: I would like to see some complementary images of land uses in the case study, for the international readers that are not from China. I suggest to add in the conclusions how this spatial methods can be integrated in a national land policy, especially related to environmental conservation. How this spatial analysis and tools can be socialized into community?
Response #1: We appreciate the expert's responsible advice. I am very grateful for your interest in these points. We have added related descriptions in “Discussion” and “Conclusion”.
Finally, the authors are grateful to the Editor and Evaluators/Reviewers for allowing us to resubmit a revised copy of the manuscript. We have revised the paper carefully according to all the comments of the reviewers to the best of our ability. We hope that the revised manuscript is acceptable to the Editor/Reviewers for publication in the journal of Science of The Total Environment.
Round 2
Reviewer 1 Report
Through this revision, the quality of the article has been greatly improved. This is the revised version, please correct to the final version and then publish.
Reviewer 2 Report
There are no comments.